# Reduction in malaria burden following the introduction of indoor residual spraying in areas protected by long-lasting insecticidal nets in Western Kenya, 2016–2018

**Diba Dulacha** [1] *, **Vincent Were** [2], **Elvis Oyugi** [1], **Rebecca Kiptui** [3], **Maurice Owiny** [1], **Waqo Boru** [1], **Zeinab Gura** [1], **Robert T. Perry** [4]

**1** Field Epidemiology and Laboratory Training Program, Ministry of Health, Nairobi, Kenya, **2** The U.S. Centers for Disease Control and Prevention, Nairobi, Kenya, **3** National Malaria Control Program, Ministry of Health, Nairobi, Kenya, **4** The U.S. President's Malaria Initiative-Kenya, Malaria Branch, Division of Parasitic Diseases and Malaria, Centre for Global Health, US Centers for Disease Control and Prevention, Atlanta, GA, United States of America

* diba8088@gmail.com

**Data Availability Statement:** The dataset used for this analysis is available from the corresponding

## Abstract

### Background

Long-lasting insecticidal nets (LLINs) and indoor residual spraying (IRS) are the main malaria vector control measures deployed in Kenya. Widespread pyrethroid resistance among the primary vectors in Western Kenya has necessitated the re-introduction of IRS using an organophosphate insecticide, pirimiphos-methyl (Actellic® 300CS), as a pyrethroid resistance management strategy. Evaluation of the effectiveness of the combined use of non-pyrethroid IRS and LLINs has yielded varied results. We aimed to evaluate the effect of non-pyrethroid IRS and LLINs on malaria indicators in a high malaria transmission area.

### Methods

We reviewed records and tallied monthly aggregate of outpatient department (OPD) attendance, suspected malaria cases, those tested for malaria and those testing positive for malaria at two health facilities, one from Nyatike, an intervention sub-county, and one from Suba, a comparison sub-county, both located in Western Kenya, from February 1, 2016, through March 31, 2018. The first round of IRS was conducted in February–March 2017 in Nyatike sub-county and the second round one year later in both Nyatike and Suba sub-counties. The mass distribution of LLINs has been conducted in both locations. We performed descriptive analysis and estimated the effect of the interventions and temporal changes of malaria indicators using Poisson regression for a period before and after the first round of IRS.

### Results

A higher reduction in the intervention area in total OPD, the proportion of OPD visits due to suspected malaria, testing positivity rate and annual malaria incidences were observed

author and the data set has been uploaded as a supporting information.

**Funding:** The author(s) received no specific funding for this work.

**Competing interests:** The authors have declared that no competing interests exist.

except for the total OPD visits among the under 5 children (59% decrease observed in the comparison area vs 33% decrease in the intervention area, net change -27%, P <0.001). The percentage decline in annual malaria incidence observed in the intervention area was more than twice the observed percentage decline in the comparison area across all the age groups. A marked decline in the monthly testing positivity rate (TPR) was noticed in the intervention area, while no major changes were observed in the comparison area. The monthly TPR reduced from 46% in February 2016 to 11% in February 2018, representing a 76% absolute decrease in TPR among all ages (RR = 0.24, 95% CI 0.12–0.46). In the comparison area, TPR was 16% in both February 2016 and February 2018 (RR = 1.0, 95% CI 0.52–2.09). A month-by-month comparison revealed lower TPR in Year 2 compared to Year 1 in the intervention area for most of the one year after the introduction of the IRS.

## Conclusions

Our findings demonstrated a reduced malaria burden among populations protected by both non-pyrethroid IRS and LLINs implying a possible additional benefit afforded by the combined intervention in the malaria-endemic zone.

## Introduction

Malaria, caused by infection with parasites of the genus *Plasmodium*, is a global public health problem with an estimated 228 million cases and 405,000 deaths reported in 2018 [1]. The prevalence of malaria found through screening during household surveys varies widely across different malaria epidemiological zones in Kenya. In the malaria-endemic counties in Western Kenya, the prevalence was estimated to be 27% in children under five years of age in 2015 [2]. Malaria and its complications, like anemia, were identified as the most common causes of childhood mortality in Western Kenya between 2003 and 2009 [3].

Long-lasting insecticidal nets (LLINs) and indoor residual spraying (IRS) are the main vector control measures recommended by the WHO [4]. Insecticide-treated nets (ITNs) are associated with a significant reduction in malaria incidence rates of 50% and malaria mortality rates of 55% among children under five years of age in Sub-Saharan Africa [5, 6]. Various approaches have been used in Kenya to distribute pyrethroid-impregnated nets, including free mass net distribution campaigns, designed to reach universal coverage, and routine distribution, targeting pregnant women and children under one year of age in endemic and epidemic-prone areas. These efforts have led to an improvement in the proportion of households nationwide with one or more nets increasing from 44% in 2010 to 63% in 2015 [7]. Small-area analyses combining Demographic Health Survey (DHS) 2014 and Kenya Malaria Indicator Survey (KMIS) 2015 estimated that the proportion of households that attained universal coverage with LLINs, defined as one LLIN for every two people in the household, was 32% in Migori County where the intervention area (Nyatike sub-county) is located and 39% in Homa Bay County where the comparison area (Suba sub-county) is located [8]. These adjoining counties, Migori and Homa Bay, are located in the malaria-endemic zone of Western Kenya around Lake Victoria, where there is intense malaria transmission throughout the year with annual entomological inoculation rates of 30–100 [2].

IRS has also been demonstrated to reduce morbidity and mortality, although data on the effectiveness of the IRS are not as comprehensive as those on insecticide-treated nets (ITNs)

[7]. In Kenya, pyrethroid-based IRS activities were stopped in 2013 when widespread pyrethroid resistance was reported among the primary vectors [9, 10]. IRS was re-introduced in February 2017 in Migori County using an organophosphate insecticide, pirimiphos-methyl (Actellic® 300CS), as part of a strategy to manage insecticide resistance [11]. WHO recommends the use of a different class of insecticide for IRS from one used in nets to reduce the development of resistance to and protect pyrethroids, for instance, the use of carbamate or organophosphate insecticide for IRS in areas where pyrethroid-impregnated LLINs are distributed [10, 11].

Kenya has conducted three free mass net distribution campaigns; in 2011–2012, 2014–2015, and 2017–2018. The net distribution campaigns were conducted in the Lake malaria-endemic counties in June 2011, June 2014 and June 2017 by the National Malaria Control Program with support from Global Fund, US President's Malaria Initiative (PMI) and other partners. In addition, free LLINs are given to pregnant women attending their first antenatal clinic and to children under one year of age attending their first visit to a vaccination clinic. The proportion of the population who reported sleeping under an LLIN was 51% in Migori County and 53% in Homa Bay County [2]. The non-pyrethroid IRS campaign in 2017 was conducted using an organophosphate, pirimiphos-methyl (Actellic® 300CS) in six of the eight sub-counties of Migori in February and March 2017. A household survey was done one to two months after the campaign estimated that 83% of eligible structures were sprayed during the first round of the IRS [12]. The second round of non-pyrethroid IRS using the same insecticide was done in the same six sub-counties of Migori County and in all sub-counties of Homa Bay County in February and March 2018. The IRS campaigns were implemented by the National Malaria Control Program with support from PMI.

The combined use of LLINs and IRS has been shown in field studies to have a more significant impact than either of the interventions used alone [13, 14]. A systematic review of studies examining the impact of IRS on key malariological parameters found enhanced protection among populations using both ITNs and IRS compared to IRS alone [7]. However, other studies have not shown any additional advantage with the combination of two vector control interventions [15]. The recent expansion of the IRS has led to the need for epidemiological data describing its impact in malaria-endemic areas. To better understand the added benefit of IRS; we compared the malaria burden at two health centers, one in an area where IRS was conducted and one in an area where it was not conducted.

## Methods and materials

### Study design

This study was a quasi-experimental design utilizing a retrospective review of medical and laboratory records for pre-intervention and post-intervention periods to compare changes in malaria indicators between intervention and comparison areas. The malaria indicators of interest were the number of confirmed outpatient malaria cases per 1000 persons per month, the proportion of suspected malaria among all cases seen in the Outpatient Department (OPD), the testing rate, and the test positivity rate. The intervention was the combination of non-pyrethroid IRS (the first round) and LLINs, which took place in the Nyatike sub-County (intervention area) in Migori County in 2017. The comparison area (Suba sub-County) received LLINs alone when the intervention area received non-pyrethroid IRS (first round) in addition to the LLINs. The comparison area was chosen based on geographical proximity and similarity of climatic conditions to the intervention area. The pre-intervention period (year 1) was from February 1, 2016, to January 31, 2017, and the post-intervention period (year 2) was from April 1, 2017, to March 31, 2018. One facility from the Nyatike sub-county where the

catchment population received the first round of non-pyrethroid IRS was selected as the intervention site, and another facility from Suba sub-county where the population did not receive the IRS during the first-round campaign was selected as the comparison site.

## Study site and vector control interventions

The two health facilities were Karungu Sub-County Hospital in Nyatike sub-county in Migori County, where the combination of non-pyrethroid IRS (first round) and LLINs took place and Suba Sub-County Hospital in Suba sub-county in Homa Bay County where the first round of non-pyrethroid IRS campaign was not done "Fig 1". Malaria is perennial in these counties, with two major peaks that usually follow the two annual rainy seasons of March/April/May and September/October/November [16].

## Data collection

We reviewed outpatient (MoH 405 A, 405 B) and laboratory (MoH 706) registers and tallied monthly aggregate numbers of all outpatient department (OPD) visits, suspected malaria cases, patients tested for malaria with either microscopy or rapid diagnostic test (RDT), and patients who tested positive for malaria, disaggregated into two age categories—younger than five years old and those aged five years and above. The OPD registers included information on each person attending the OPD, clinical diagnosis, tests that were requested, and the treatment given. The laboratory registers provide for each person tested, tests performed, and the results. Data review covered one year before the first round of IRS (from February 1, 2016, through January 31, 2017) and one year after the first round of IRS (from April 1, 2017, through March 31, 2018) for both facilities.

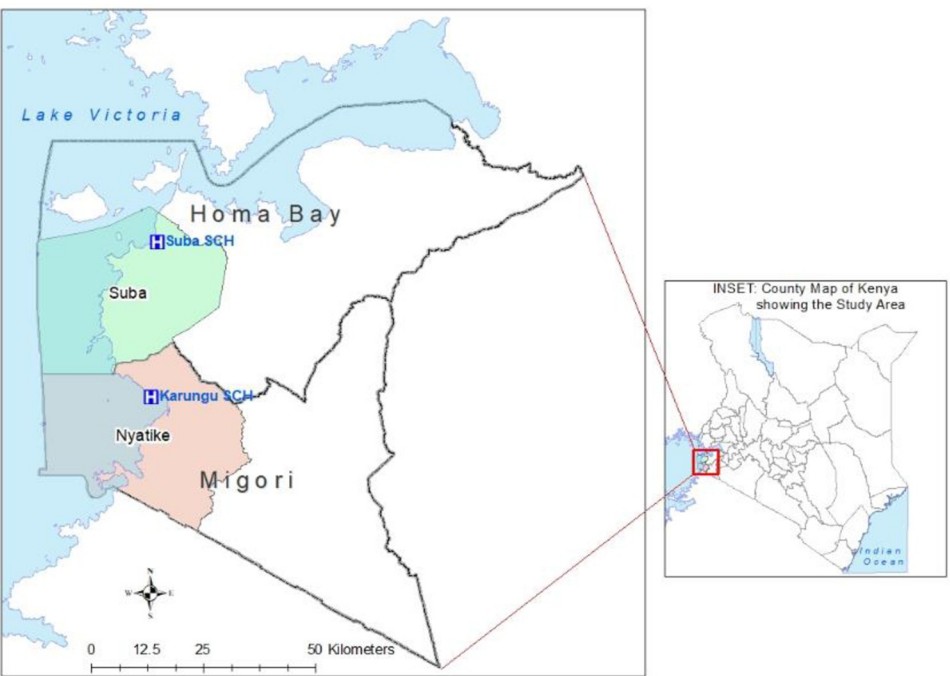

**Fig 1. A map of the intervention area (Nyatike sub-county) and the comparison area (Suba sub-county) in Kenya, 2018.**

## Data management and analysis

We double-entered data into MS Excel for cleaning and analyzed the data using STATA (STATA Corp., College Station, TX, USA). For the months with missing variables, we imputed using the averaging method where the missing values were generated using averages of the values of the preceding month and the month that follows the month with a missing value.

Annual malaria incidence was the number of confirmed cases per 1,000 populations using the mid-year estimated catchment population of each facility. The test positivity rate was the ratio of slides or RDT tests done that tested positive for malaria over the number of patients tested by microscopy or RDT. We also calculated the proportion of all outpatient department (OPD) visits with a diagnosis of suspected malaria as the number of patients diagnosed with suspected malaria divided by the total number of all OPD visits.

We created two categorical variables to use in regression models, one for spray status, to allow comparisons between the intervention area and the comparison area, and another for the time, to allow comparisons between the period before IRS (February 1, 2016, to January 31, 2017), labeled as "Year 1", and the period after IRS in Nyatike Sub-county (April 1, 2017, to March 31, 2018), labeled as "Year 2". We used the time variable to estimate the differences in malaria indicators between Year 1 and Year 2 and the spray status variable to compare changes in malaria indicators between the facility in the area where the first round of IRS was done and the facility in the area where it was not done. We applied the difference-in-differences (DID) approach [17, 18], to estimate the difference between Year 1 and Year 2 for the proportion of OPD visits due to suspected malaria and testing positivity rates, and differences between percentage change for total OPD visits and annual malaria incidences. We used a mixed-effect Poisson regression model to measure the effect of the intervention on the malaria indicators where the coefficient of the interaction term between spray status variable and period of study (year 1 vs. year 2) represented the net of effect of IRS. The comparison of monthly test positivity rates was made by calculating the relative rate (RR) using Poisson regression to assess the presence of significant differences in temporal trends of the indicators over the period of interest. We calculated the RR of monthly testing positivity rates (TPRs) without any adjustments for other variables for month-to-month comparison. For the Poisson regression model, the relative rate (RR) of $<1$ was considered protective, and the p-value of $<0.05$ was statistically significant.

## Ethical consideration

Moi University Institution for Research and Ethics Committee (IREC) approved this study (Approval Number: 0002048), and the Health Departments of Migori and Homa Bay Counties gave consent to access the health facilities and the records. The variables collected were fully anonymized monthly summaries from health records. The study underwent human subject review at the U.S. Centers for Disease Control and Prevention and was approved as a program evaluation activity that does not require human subject research review.

## Results

The combination of the non-pyrethroid IRS and LLINs in the intervention was associated with a higher percentage decrease for all indicators except the OPD visits among the under 5 children where 59% decrease was observed in the comparison area while a 33% decrease was observed in the intervention area (net change -27%, P $<0.001$). A negative percentage change (-4%) in the total OPD visits was observed among the 5-and-over population as a slight increase in total OPD visits was observed in Year 2.

**Table 1. Changes in annual malaria indicators in the intervention area and comparison area before and after the introduction of the first round of IRS, 2016–2017.**

| Malaria indicators | Intervention area (IRS +LLIN) | | | Comparison area (LLIN alone) | | | Net change | |
|---|---|---|---|---|---|---|---|---|
| | Year 1 | Year 2 | Change (A) | Year 1 | Year 2 | Change (B) | A-B | P-value |
| Total OPD visits* | | | % change | | | % change | Int-comp | |
| All ages | 12460 | 6948 | 44% | 19823 | 15160 | 24% | 21% | <0.001 |
| <5 | 2741 | 1848 | 33% | 8621 | 3502 | 59% | -27% | <0.001 |
| ≥5 | 9719 | 5100 | 48% | 11202 | 11658 | -4% | 52% | <0.001 |
| Suspected malaria cases† (% of OPD visits) | | | Diff % | | | Diff % | DiD | |
| All ages | 3966 (32) | 746 (11) | -21 | 7284 (37) | 4780 (32) | -5.2 | -16 | <0.001 |
| <5 | 1026 (37) | 184 (10) | -27 | 2607 (30) | 801 (23) | -7.3 | -20 | <0.001 |
| ≥5 | 2940 (30) | 562 (11) | -19 | 4677 (42) | 3979 (34) | -7.7 | -12 | <0.001 |
| Tested positive‡ (% of tested) | | | Diff % | | | Diff % | DiD | |
| All ages | 3847 (39) | 436 (14) | -25 | 2929 (19) | 1147 (16) | -3.4 | -21 | <0.001 |
| <5 | 1144 (27) | 172 (16) | -11 | 1331 (23) | 469 (17) | -5.6 | -6 | 0.006 |
| ≥5 | 2703 (47) | 264 (13) | -34 | 1598 (17) | 678 (15) | -1.8 | -32 | <0.001 |
| Malaria incidence/1000§ | | | % change | | | % change | Int-comp | |
| All ages | 360 | 38 | 89% | 131 | 78 | 40% | 49% | <0.001 |
| <5 | 552 | 78 | 86% | 360 | 194 | 46% | 40% | <0.001 |
| ≥5 | 314 | 29 | 91% | 85 | 55 | 35% | 56% | <0.001 |

OPD = Outpatient department; Int-change in intervention area; Comp-change in comparison area; DiD-difference-in-differences

* All outpatient visits as recorded in the OPD registers

† Proportions of all outpatient visits contributed by suspected malaria cases

‡ Number of samples tested positive for malaria divided by the total number of malaria tests done

§ Annual malaria incidence rates using the mid-year population estimates for the two facilities as the denominators and expressed as per 1,000 populations.

The greatest net decline in the proportion of OPD attendance attributed to suspected malaria cases and TPR was observed among the under-five children (-27% intervention area vs -7.3% comparison area, net change -20%, P < 0.001) and among the five-and-over population (-34% intervention area vs. -1.8% comparison area, net change -23%, P <0.001) respectively.

The percentage decline in malaria incidences observed in the intervention area was more than twice the observed percentage decline in the comparison area across all the age groups "Table 1". The missing records for the number of malaria tests performed and the number of confirmed malaria cases for February 2016, February 2017, and December 2017 were computed for the comparison area (Table 1).

Before the introduction of the non-pyrethroid IRS in the intervention area, overall monthly TPR was higher in the intervention area than in the comparison area. A marked decline in the monthly TPR was observed in the intervention area while no major changes are observed in the comparison area upon introduction of the non-pyrethroid IRS in the intervention area. The monthly TPR reduced from 46% in February 2016 (start of review period) to 11% in February 2018 (end of review period), representing a 76% absolute decrease in TPR among all ages (RR = 0.24, 95% CI 0.12–0.46). In the comparison area, TPR was 16% in both February 2016 and February 2018 (RR = 1.0, 95% CI 0.52–2.09). A month-by-month comparison revealed that the TPR in Year 2 remained lower than in Year 1 in the intervention area for most of the one year after the introduction of the IRS. In comparison area, the overall TPR remained relatively stable in Year 1 and Year 2. An unexplained spike was observed in the comparison area in March 2018 in Year 2 "Fig 2".

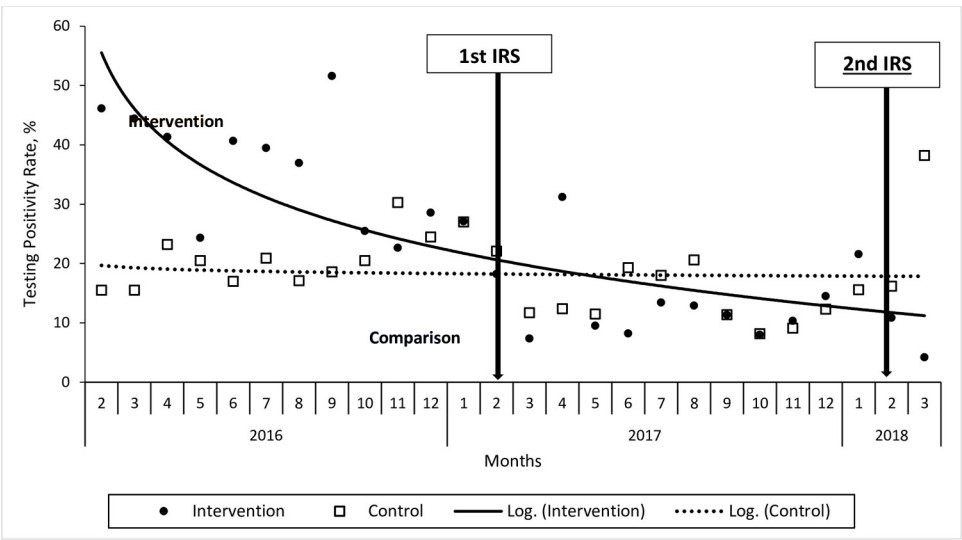

**Fig 2. Trends of monthly malaria test positivity rates among all ages in the intervention area and comparison area, 2016–2018.** The solid line represents the line of best fit for the intervention area while the dotted line represents the line of best fit for the comparison area. The lines are plotted on a logarithmic scale. Solid circles mark the data points for the intervention area while the open squares marked the data points for the comparison area. The regression coefficient for the intervention area was y = -13.6ln(x) + 55.536 and y = -0.566ln(x) + 19.688 for the comparison area.

## Discussion

Our retrospective review of medical records found a marked reduction in malaria incidence, TPR, and proportion of OPD visits due to suspected malaria in the catchment area of a hospital in a sub-county where IRS was done, compared to the catchment area of a hospital in a sub-county where IRS was not done. Both areas are malaria-endemic, had moderate rates of LLIN possession, experienced similar climatic and rainfall patterns and are representative of the Lakes endemic counties. The high malaria transmission seasons in Year 1 and Year 2 were included in the review period and the comparison for key malariological indicators was done between two full calendar years. Our findings suggest that adding IRS with a non-pyrethroid insecticide to LLINs impregnated with pyrethroid insecticides is effective in further reducing the malaria burden in an area of widespread pyrethroid resistance. These findings add to a growing body of evidence showing additional benefits afforded by a combination of IRS and LLINs using different insecticides in malaria-endemic areas and the significance of managing insecticide resistance.

A similar study utilizing enhanced routine surveillance data at an outpatient facility in Apac District in Uganda demonstrated a dramatic decline in malaria morbidity after initiation of IRS with bendiocarb and resurgence in malaria cases three months after discontinuation of IRS [19]. In addition, a pre-post comparison study done in Tororo District in Uganda district using secondary data from health facilities for the period between 2013 and 2015 noted a significant reduction in the incidence of malaria among < 5-year-old children from 130 to 100 cases per in 2014 when LLINs were used alone and a further significant decline to 45 cases per 1000 in 2015 when IRS was combined with LLINs [20]. The presence of high-level pyrethroid resistance among the primary vectors in multiple parts of Uganda, including both Apac and Tororo, has been demonstrated [21, 22]. A non-randomized prospective study in western Kenya comparing the impact of combined use of ITNs and IRS against the use of ITNs alone also showed a lower incidence of *P. falciparum* (18 per 100 persons-years at risk) among the

group with both ITNs and IRS than the group with ITNs only (44 per 100 persons-year at risk) with an adjusted rate ratio of 0.41 (95% CI 0.31–0.56) [14]. Another study done in 2008–2009 in Rachuonyo and Nyando areas in western Kenya, where prevalence was measured by repeated household surveys of randomly-selected households visited every month, showed lower malaria prevalence in a district with pyrethroid IRS (lambda-cyhalothrin in year 1 and, alphacypermethrin in year 2) compared with a neighboring district where only ITNs were provided (6.4% vs 16.7%, OR = 0.36, 95%CI 0.22–0.59) [23]. A review of data from six countries examining the effect of combining IRS and LLINs found mixed results, with possibly additional benefits from the combination in a setting with low-medium LLIN usage [24]. IRS implemented targeting high coverage of more than 85% of eligible structures, is associated with a rapid reduction in vector population and affords protection to the community members not sleeping under mosquito nets and additional protection to those who sleep under LLINs/ITNs, explaining the more significant reduction among those in houses covered by IRS who are also using LLINs. While LLINs can potentially have a similar effect on malaria vector populations its efficacy is affected by failure to achieve universal coverage despite multiple mosquito net distributions campaigns and the widespread pyrethroid resistance.

Other studies have shown protective effects of IRS and ITNS used together but failed to demonstrate any difference between the use of the combination and either intervention alone. Protopopoff and colleagues found in Burundi that despite reduced vector densities inside houses secondary to the use of ITNs and IRS, the overall transmission of malaria was not significantly reduced relative to when either of the two methods was used alone. However, this result may reflect more the choice of the control site [25]. In rural Gambia with moderate seasonal malaria transmission and high LLINs coverage, the addition of IRS using DDT as the insecticide did not result in a significant reduction in the levels of clinical malaria compared to the control group where LLINs were deployed alone. The failure to observe a significant reduction in malaria incidences in this study is attributed to the high LLINs coverage, of 83–95%, among the children in the cohort whose households received LLINs as part of the study that led to a reduced number of blood-feeding mosquitoes settling on the walls [26].

Our study had some limitations; gaps in the quality of the routine facility data may have affected our analysis. The ratio of cases tested in the laboratory to cases seen in the outpatient department was consistently higher than one, likely a result of our inability to restrict the data collection to only outpatients and exclude cases coming from other parts of the hospital like the antenatal clinic or inpatient wards. In addition, some RDTs done outside the laboratory or by non-laboratory personnel in different parts of the facilities may not have been recorded in the laboratory registers. We calculated incidence based on the health facility catchment population, but clients may have come from areas outside the catchment area as the selected facilities are the major referral health facilities in the two sub-counties. The limited number of health facilities included for data collection limits our ability to generalize our findings to the whole sub-county or county. The results are also subject to selection bias for testing and the accuracy of laboratory testing. Further, the inability to exclude the tests done on the inpatient cases meant that a certain percentage of our positive cases could have been the admitted cases. The other limitation was the lack of data on climatic and environmental factors that affect malaria risk and transmission. However, these issues could have affected both the intervention and comparison sites, thus reducing the effect of these biases on the general findings.

For at least up to the last malaria indicator survey in 2015, Suba and Nyatike had inadequate coverage with LLINs–and since one needs to possess a net to use it, these areas also had reduced rates of net use. Pyrethroid resistance in mosquito vectors may have blunted the effectiveness of pyrethroid-based LLINs. This analysis showed that adding IRS with a non-pyrethroid insecticide in Nyatike seemed to reduce the incidence, compared to no real change in

Suba, at least at the sub-county hospital OPD. Expansion of IRS to other malaria-endemic counties, along with considering the use of newer nets to achieve universal coverage, could reduce the malaria burden further.

## Supporting information

**S1 Dataset. Migor Homa Bay IRS LLINs data set.**
(XLSX)

## Acknowledgments

We acknowledge Migori and Homa Bay counties and their staff for their cooperation and assistance during data collection. We also acknowledge Kenya FELTP, and Moi University for their technical support.

## Author Contributions

**Conceptualization:** Diba Dulacha, Elvis Oyugi, Rebecca Kiptui, Maurice Owiny, Waqo Boru, Zeinab Gura, Robert T. Perry.

**Data curation:** Diba Dulacha.

**Formal analysis:** Diba Dulacha, Vincent Were.

**Funding acquisition:** Rebecca Kiptui, Maurice Owiny, Waqo Boru, Zeinab Gura, Robert T. Perry.

**Investigation:** Diba Dulacha.

**Methodology:** Diba Dulacha, Zeinab Gura, Robert T. Perry.

**Project administration:** Zeinab Gura.

**Supervision:** Diba Dulacha, Rebecca Kiptui, Zeinab Gura, Robert T. Perry.

**Validation:** Diba Dulacha.

**Visualization:** Zeinab Gura, Robert T. Perry.

**Writing – original draft:** Diba Dulacha, Robert T. Perry.

**Writing – review & editing:** Diba Dulacha, Elvis Oyugi, Robert T. Perry.

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
