## [Decision Letter · Decision Letter 0]

28 Jan 2022

PONE-D-21-39990Reduction in malaria burden following the introduction of indoor residual spraying in areas protected by long lasting insecticidal nets in Western Kenya, 2016 – 2018PLOS ONE

Dear Dr. Dulacha,

Thank you for submitting your manuscript to PLOS ONE. After careful consideration, we feel that it has merit but does not fully meet PLOS ONE’s publication criteria as it currently stands. Therefore, we invite you to submit a revised version of the manuscript that addresses the points raised during the review process. Please address the issues raised by the reviewer (you can ignore the issues about line numbers if this is hard for you to address). 

We look forward to receiving your revised manuscript.

Kind regards,

Jan Rychtar

Academic Editor

PLOS ONE

Journal Requirements:

"This work was funded by Kenya Field Epidemiology Training Program with support from the US President's Malaria Initiative (PMI) and Centers for Disease Control and Prevention (CDC). The lead author was a resident with the program and was supported to conduct this research through this funds."

The senior author, who is a US CDCD employee functioned as a supervisor for the lead author during his residency and study. The supervisor supported in designing, analysis and reviewing the manuscript..  

5. We noted in your submission details that a portion of your manuscript may have been presented or published elsewhere. [This study was part of a thesis project and thesis report is available on Moi University website.] Please clarify whether this [conference proceeding or publication] was peer-reviewed and formally published. If this work was previously peer-reviewed and published, in the cover letter please provide the reason that this work does not constitute dual publication and should be included in the current manuscript.

7. We note that Figure 1 in your submission contain map image which may be copyrighted. All PLOS content is published under the Creative Commons Attribution License (CC BY 4.0), which means that the manuscript, images, and Supporting Information files will be freely available online, and any third party is permitted to access, download, copy, distribute, and use these materials in any way, even commercially, with proper attribution. For these reasons, we cannot publish previously copyrighted maps or satellite images created using proprietary data, such as Google software (Google Maps, Street View, and Earth). For more information, see our copyright guidelines: http://journals.plos.org/plosone/s/licenses-and-copyright.

Additional Editor Comments:

This is a good manuscript that will be acceptable for publication once the authors address the minor revisions raised by the reviewer.

Reviewers' comments:

Reviewer's Responses to Questions

**Comments to the Author**

1. Is the manuscript technically sound, and do the data support the conclusions?

Reviewer #1: Yes

 ********** 

2. Has the statistical analysis been performed appropriately and rigorously? 

Reviewer #1: Yes

3. Have the authors made all data underlying the findings in their manuscript fully available?

Reviewer #1: No

4. Is the manuscript presented in an intelligible fashion and written in standard English?

Reviewer #1: Yes

5. Review Comments to the Author

Reviewer #1: This article reports on large-scale vector control measures conducted in a malaria-endemic areas of Western Kenya around Lake Victoria from 2016-2018. It aims to evaluate the effect of pyrethroid LLINs with and without non-pyrethroid IRS on malaria in a high malaria transmission area, annual entomological inoculation rates of 97 30 – 100 [16].

This study uses data of the recent large-scale LLIN/IRS undertaken in this subregion and as such these results deserves publication in a journal like PLOSone but some changes are required.

1. This study is based on data from two sub-counties, Nyatike, and 49 Suba, located on the shore of Lake Victoria, Western Kenya in adjacent counties, Migori and Homa 95 Bay, where there 96 is intense malaria transmission throughout the year. The interventions reported on were conducted from February 1, 2016 to 50 March 31, 2018.

--Please add a comment on how representative the data from Nyatike and Suba are of data from this subregion.

2. The authors state that “Kenya conducted three free mass net distribution campaigns in 2011–2012, 2014–2015, and 2017–2018”.

--Details of the date(s) and organisation(s) responsible for net distribution should be stated for the sub-counties that provided the data for this article.

3. IRS was conducted in February – March 2017 in Nyatike but not in Suba. One year later both Nyatike and Suba sub-counties received IRS.

--Please add details of the organisation responsible for IRS for these two sub-counties and the reason for some counties receiving IRS and not others.

4. Two health facilities, one from Nyatike and one from Suba sub-county, provided out-patient data from February 1, 2016, to March 31, 2018. The data included outpatient department attendance, suspected malaria cases, those tested for malaria and those testing positive for malaria.

--was the data collection blinded?

--was it double entered?

5. lines 177 -180. “..entered data into MS Excel for cleaning”. Excel is a spreadsheet application and not a data management programme. Was the data double entered?

6. Line 187. “Months with missing values were imputed”. Please add the frequency that months were missing.

7. The authors have omitted line numbers from Table 1 onwards.

8. Results are clearly presented.

9. The discussion could be more succinct

6. PLOS authors have the option to publish the peer review history of their article (what does this mean?). If published, this will include your full peer review and any attached files.

Reviewer #1: No

---

## [Author Response · Author response to Decision Letter 0]

12 Mar 2022

Thank you for the comments and guidance.

We have provided itemized response to each issue raised by the editor and reviewer #1 in the 'Response to Reviewers'.

---

## [Editor Report · Decision Letter 1]

28 Mar 2022

Reduction in malaria burden following the introduction of indoor residual spraying in areas protected by long-lasting insecticidal nets in Western Kenya, 2016 – 2018

PONE-D-21-39990R1

Dear Dr. Dulacha,

We’re pleased to inform you that your manuscript has been judged scientifically suitable for publication and will be formally accepted for publication once it meets all outstanding technical requirements.

Kind regards,

Jan Rychtar

Academic Editor

PLOS ONE

Additional Editor Comments (optional):

The authors addressed the comments raised in the previous round of reviews and the manuscript can now be accepted for the publication.
---

## [Editor Report · Acceptance letter]

11 Apr 2022

PONE-D-21-39990R1 

Reduction in malaria burden following the introduction of indoor residual spraying in areas protected by long-lasting insecticidal nets in Western Kenya, 2016 – 2018   

Dear Dr. Dulacha:

I'm pleased to inform you that your manuscript has been deemed suitable for publication in PLOS ONE. Congratulations! Your manuscript is now with our production department. 

Kind regards, 

on behalf of

Dr. Jan Rychtar 

Academic Editor

PLOS ONE